# FinSurvival: A Suite of Large Scale Survival Modeling Tasks from Finance

**Aaron Green**                                                GREENA12@RPI.EDU
*Department of Mathematical Sciences*
*Rensselaer Polytechnic Institute*
*Troy, NY 12180-3950, USA*

**Zihan Nie**                                                      NIEZ@RPI.EDU
*Department of Mathematical Sciences*
*Rensselaer Polytechnic Institute*
*Troy, NY 12180-3950, USA*

**Hanzhen Qin**                                                 QINH2@RPI.EDU
*Department of Computer Science*
*Rensselaer Polytechnic Institute*
*Troy, NY 12180-3950, USA*

**Oshani Seneviratne**                                         SENEVO@RPI.EDU
*Department of Computer Science*
*Rensselaer Polytechnic Institute*
*Troy, NY 12180-3950, USA*

**Kristin P. Bennett**                                         BENNEK@RPI.EDU
*Departments of Mathematical Sciences and Computer Science*
*Rensselaer Polytechnic Institute*
*Troy, NY 12180-3950, USA*

## Abstract

Survival modeling predicts the time until an event occurs and is widely used in risk analysis; for example, it's used in medicine to predict the survival of a patient based on censored data. There is a need for large-scale, realistic, and freely available datasets for benchmarking artificial intelligence (AI) survival models. In this paper, we derive a suite of 16 survival modeling tasks from publicly available transaction data generated by lending of cryptocurrencies in Decentralized Finance (DeFi). Each task was constructed using an automated pipeline based on choices of index and outcome events. For example, the model predicts the time from when a user borrows cryptocurrency coins (index event) until their first repayment (outcome event). We formulate a survival benchmark consisting of a suite of 16 survival-time prediction tasks, called FinSurvival. With over 7.5 million records, FinSurvival provides a suite of realistic financial modeling tasks that will spur future AI survival modeling research. Our evaluation indicated that these are challenging tasks that are not well addressed by existing methods. FinSurvival enables the evaluation of AI survival models applicable to traditional finance, industry, medicine, and commerce, which is currently hindered by the lack of large public datasets. Our benchmark demonstrates how AI models could assess opportunities and risks in DeFi. In the future, the FinSurvival benchmark pipeline can be used to create new benchmarks by incorporating more DeFi transactions and protocols as the use of cryptocurrency grows.

**Keywords:** survival analysis, benchmark, big data, finance, decentralized finance, defi, fintech, lending, Aave, blockchain

## 1 Introduction

Survival data, also called time-to-event data, is used to create models for how long it takes for certain events to occur. This kind of data arises in a wide range of disciplines, most notably in finance, where events of interest could include loan defaults [DUFFIE et al. (2009); Lando (1998)], bankruptcy [Shumway (2001)], or customer churn, and in medicine, where events of interest could include the recovery or death of a patient. Given the nature of these disciplines, survival datasets about these events can be difficult to obtain. These datasets tend to be sensitive or private, with many deep-learning-based survival methods being based on economic data that requires expensive paid subscriptions or medical datasets that have restricted or no availability [Ranganath et al. (2016); Miscouridou et al. (2018); Jing et al. (2019); Lee et al. (2018)]. In addition, these datasets tend to be limited in size. Many popular survival datasets such as METABRIC [Curtis et al. (2012)] or SUPPORT [Knaus et al. (1995)] have only between 1,500 and 10,000 records, and in SurvSet [Drysdale (2022)], which contains a repository of 76 survival datasets, the largest dataset has just 52,422 records, and most are much smaller. Not only do these datasets have relatively few records, but they also have very few features, most containing fewer than ten features. Beyond individual datasets such as METABRIC and SUPPORT, much of the modern survival modeling literature has relied on clinical trial repositories such as `ClinicalTrials.gov`, which contains thousands of studies reporting time-to-event outcomes across a wide range of biomedical applications. These repositories have provided the foundation for many methodological advances in survival analysis but remain challenging to use for machine-learning benchmarks due to inconsistent data formats, privacy restrictions, and limited public accessibility. As a result, nearly all widely used public survival datasets–those aggregated in resources such as SurvSet–derive from the biomedical domain, with relatively small sample sizes and few features. The prevalence of these clinical datasets has shaped the trajectory of survival-model development, leading to methods that are well suited for small, highly censored medical data but rarely evaluated on large-scale, open, and non-clinical settings.

Our work addresses this gap by introducing FinSurvival, a benchmark of comparable scope and rigor that is derived entirely from open financial transaction data. By constructing sixteen time-to-event prediction tasks from decentralized-finance (DeFi) lending protocols, FinSurvival expands the scope of survival analysis beyond traditional domains and provides a foundation for evaluating modern survival methods under real-world, high-censoring conditions with millions of observations. One prominent application of survival analysis is in the field of Omics. The data in this domain typically contains huge amounts of features ($> 4000$), but very few records ($< 100$). Paid economic survival data can contain large datasets such as Moody's Default and Recovery Database [1], which has over 850,000 records, but the cost to access the data can be prohibitively high. Given the effectiveness of deep learning models in nearly every discipline, and given these models' need for lots of training data, the existing survival datasets are too small to truly assess the capacities of state-of-the-art models. We address this gap by publishing a novel collection of survival datasets based on free, publicly accessible financial transaction data from the decentralized finance (DeFi) space that consists of 16 different time-to-event scenarios and combines

---

1. `https://www.moodys.com/sites/products/ProductAttachments/DRDDocumentationv2/DRDV2_FAQ.pdf`

Table 1: Comparison of our dataset with other publicly available survival datasets.

| Dataset | Domain | # Records | # Features | Source |
|---|---|---|---|---|
| FinSurvival | Finance | 7,698,497 | 128 | this paper |
| Melanoma | Omics | 41 | 642 | Wang et al. (2020) |
| Ovarian | Omics | 58 | 19,818 | Ganzfried et al. (2013) |
| SUPPORT | Clinical | 9,105 | 47 | Knaus et al. (1995) |
| METABRIC | Clinical | 1,980 | 9 | Curtis et al. (2012) |
| WHAS | Clinical | 1,638 | 5 | Floyd et al. (2009) |
| GBSG | Clinical | 686 | 9 | Foekens et al. (2000) |
| hdfail | Engineering | 54,422 | 6 | Monaco et al. (2018) |

7,698,497 records, averaging over 481,000 records per dataset. We show a comparison of our data size with various public survival datasets from other domains in Table 1. To the best of our knowledge, this is the first large-scale, publicly available financial survival dataset derived from DeFi transactions. Additionally, our datasets contain no personal identification information nor intellectual property and are freely available for research use.

DeFi is an emerging area within the cryptocurrency world that aims to provide financial services without the need for traditional banks. It uses blockchain technology and smart contracts to offer services like lending, borrowing, trading, and earning interest on crypto assets. One major type of DeFi application is the lending protocol. Lending protocols function similarly to banks in traditional finance, allowing users to deposit their monetary assets into a savings account and accrue some interest, as well as borrow funds from the protocol using their deposited assets as collateral. In this work, we use data from one of the leading lending protocols, Aave [Boado, Ernesto (2020)], which has more than $27 billion locked across eight different networks and 15 markets as of April 4, 2025.

In Aave, we study five key transaction types that facilitate lending and borrowing activities: deposits, borrows, repays, withdraws, and liquidations. The protocol *users* are actually cryptocurrency wallets with no identifying information. *Deposit* transactions involve users supplying a cryptocurrency to the protocol to earn interest over time while providing liquidity for borrowers. Deposits also serve as collateral for the users' loans. *Borrow* transactions allow users to take out loans against their deposited collateral, enabling them to access liquidity without selling their assets. *Repay* transactions refer to the act of paying back borrowed funds, reducing the borrower's outstanding debt and interest obligations. *Withdraw* transactions enable users to retrieve their deposited assets, provided they still meet the collateral requirements after the withdrawal. Lastly, *liquidation* transactions occur when a borrower's collateral value falls below the required threshold, triggering the sale of collateral to repay the loan and protect the protocol's solvency. The cryptocurrency

used in a transaction is referred to as the *reserve*. These transactions collectively define the fundamental financial dynamics of lending in Aave. There are some natural questions one can ask about user behaviors based on these transaction types, such as "How long do users take to repay loans after borrowing?" or "How long do users leave deposited money in their account before withdrawing it?" We build distinct survival datasets to model these types of questions.

In this paper, we collected raw Aave user transaction data from TheGraph[2] and created a pipeline for transforming the transaction data into survival data. This process involves selecting one of the transaction types as an index event and another transaction type as an outcome event, collating transactions based on the user and coin (i.e., reserve) used in the transaction, and computing the time elapsed between the index and outcome events. We use this process to create 16 distinct survival datasets, which we explain in detail in Section 2. Each of these datasets corresponds to a different user behavior pattern in Aave and enables meaningful survival analyses. For each dataset, we used domain knowledge to derive 128 features describing the transaction and prior account history for the index event. To demonstrate the utility of our datasets, we define the benchmark tasks of Time-to-Event Prediction, which involves estimating the expected time until an outcome event occurs. In addition to its scale and openness, our FinSurvival dataset fills a critical gap in the evaluation of survival models under real-world conditions of high-censoring and structured financial data. FinSurvival offers structured, high-dimensional data derived from real financial behavior with a mean censoring rate exceeding 80% across its 16 datasets. Thus, this dataset provides the machine learning community with a much-needed testbed for developing and evaluating models that must perform reliably when event signals are rare, covariates are rich, and data is complex–conditions that are increasingly relevant in financial domains.

**Contributions:** Our paper makes the following contributions:

- **Release of novel, large survival datasets**: We created and released a collection of 16 large-scale survival datasets derived from real financial transaction data. The dataset is available on Zenodo. [3]

- **Benchmarking results:** We create a time-to-event task for each of these datasets, then benchmark several models' performance for these tasks.

- **Open-source code:** All code written to reproduce the content of this paper is published in a GitHub repository [4]. This includes code to transform raw transaction data (along with a sample of raw transaction data) into survival data, code to compute data statistics in this paper, and code to reproduce the experiments.

The rest of this paper is organized as follows. In Section 2 we describe our datasets in detail and explain how they were converted into train and test sets for experiments. In Section 3 we explain how we set up the survival prediction task for each dataset and provide benchmark results for several survival methods on this task. Finally, we conclude with a discussion of the overall results in Section 4 and how this work can be extended in the future in Section 5.

---

2. `thegraph.com`

3. `https://zenodo.org/records/17352978`

4. `https://github.com/Large-Transaction-Models/DMLR_DeFi_Survival_Benchmark`

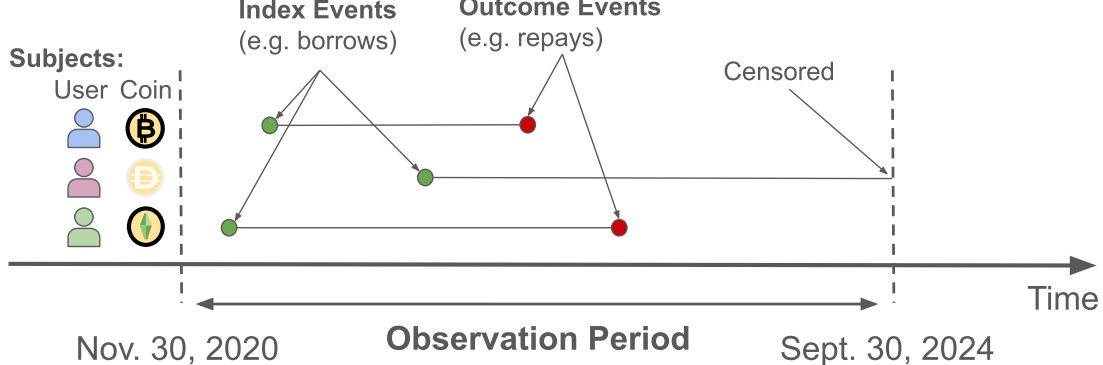

Figure 1: The idea behind survival data. One or more subject types are selected and observed over a given observation period. Activity is monitored, waiting for an index event to trigger the start of a record. A chosen outcome event marks the end of the record, or the record is censored at the end of the observation period.

## 2 Dataset Description

### 2.1 Overview of Survival Data

Survival analysis data are typically collected by identifying a cohort of subjects and recording the time until an event of interest, known as the "outcome event," occurs. The starting point for measuring time is often referred to as the "index event," which can be an initial diagnosis, treatment commencement, or any other significant starting point relevant to the study. Data collection involves tracking subjects over a specified period, noting whether and when the outcome event, such as death, relapse, or recovery, happens. Additionally, for those subjects who do not experience the outcome event within the observation period, their data are considered censored at the last point of follow-up, thereby accounting for incomplete observations. This method allows researchers to analyze the time-to-event data, accommodating both observed and censored cases, to derive meaningful insights into the factors influencing survival times. See Figure 1 for a visualization of this idea in the context of our data.

Our suite of survival datasets is all created from raw transaction data from the DeFi lending protocol Aave, specifically the Aave V2 Ethereum protocol. This data was acquired from The Graph[5]. We built a pipeline to convert the raw transaction data into survival data, using this pipeline to create 16 survival datasets. These datasets were built by separately treating each transaction type (except for liquidations) as index events, and subsequently each of the other transaction types as a possible outcome event. Liquidations, which roughly correspond to partial defaults on a borrowing transaction, are quite rare. This produces four possible index events, each with four possible outcome events. One notable feature of our data is that, because we were able to collect every transaction since Aave launched, we have no left-censored records. Every record has an associated index event. For an example of

---

5. `thegraph.com`

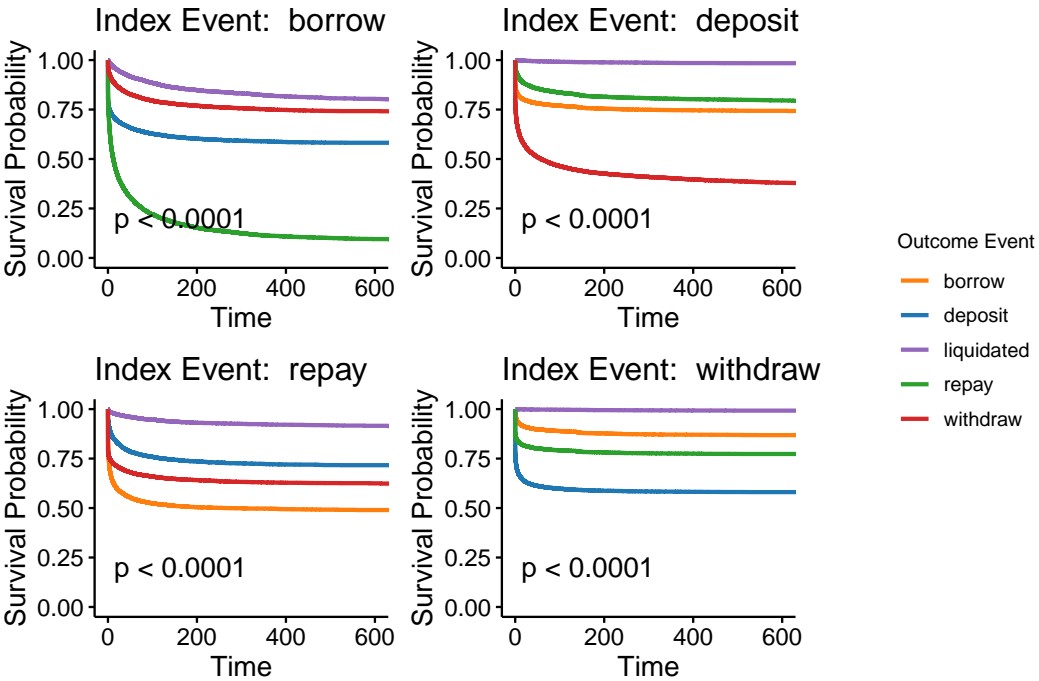

Figure 2: Kaplan–Meier curves for all index event and outcome event combinations. Each plot contains four curves representing a single index event and the four outcome events possible for that index event. These curves show that there is a variety of behaviors across our different datasets. The p-values are obtained from a log-rank test comparing survival distributions across the four outcome-event types and indicate strong statistically significant differences in outcome event behaviors for each index event.

what one of our survival datasets looks like, see Table 2. For an overview of all 16 datasets, see Table 4. We also show Kaplan–Meier survival curves for all 16 datasets in Section 2.1 to show that these datasets represent different patterns of behavior.

We also include 128 features for this survival data based on prior publications and domain knowledge [Green et al. (2022, 2023)]. The raw data for each transaction contains 22 features, most of which can be kept as features for each record. These include features like which coin was used in a transaction (reserve), how much money was involved in a transaction, the lending pool involved, etc. Different transaction types have slightly different inherent features, and if a feature is irrelevant for a particular transaction type, we leave it as NA. The exact features are listed in Table 9 in Appendix B.4.

On top of these features, we engineered larger sets of features following three major themes. We built 19 features to represent the temporal aspect of transactions in different ways, encoding the date and time of each transaction into cyclic representations based on the day of the month, day of the week, day of the quarter, etc. We built 45 user-history features

that represent up to the time of a given transaction a summary of the user's transaction history, such as how much money they have spent on each transaction type, how frequently they have made transactions, what coins they use most often, etc. Similarly, we built 40 features that represent a summarized history of the market as a whole up to the time of the transaction, such as how much of a specific coin in a transaction has been borrowed or deposited up to that point. A more detailed overview of these features can be found in Appendix B.4.

## 2.2 Feature Evaluation via Univariate Survival Analysis

Because the features in FinSurvival are manually engineered and domain-specific, we performed a lightweight feature-level evaluation to verify that each feature exhibits nontrivial association with survival outcomes. This analysis is intended as a sanity check on feature relevance rather than as a comprehensive feature-importance study or a substitute for model-specific interpretability analyses. Our goal is to ensure that the benchmark feature set does not contain inert or purely noisy covariates prior to model evaluation.

For each of the sixteen index-outcome event pairs, we fit univariate Cox proportional hazards models of the form $\lambda(t|X) = \lambda_0(t)\exp(\beta X)$ using each feature individually as a covariate. For each feature-task combination, we recorded the estimated hazard ratio and corresponding Wald test p-value. To account for multiple feature-wise hypothesis tests within a given task, p-values were adjusted using the Benjamini-Hochberg procedure [Benjamini and Hochberg (1995)], yielding q-values that control the false discovery rate at the task level.

We considered a feature to exhibit meaningful survival signal for a task if it satisfied two criteria simultaneously: (i) a statistically detectable association with event timing, defined by having a q-value below 0.005, and (ii) a non-negligible effect size, defined as an absolute log-hazard ratio of at least 0.1, corresponding to an approximate 10% or greater multiplicative change in hazard per unit increase in the feature. Under these conservative criteria, every feature in the benchmark demonstrated detectable survival signal for at least one task. This result supports the inclusion of the full feature set in the benchmark while preserving task heterogeneity: most features are informative only for subsets of index-outcome pairs, reflecting the diverse behavioral dynamics captured by the dataset. A summarized breakdown of these feature-level results is provided in Appendix B.5.

## 2.3 Train/Test Split

We had to split the data temporally into a training and a testing set to run experiments on this data. To do this, we chose a cutoff date of July 1, 2022. The training set includes data before this cutoff date, and the testing set includes data afterwards. This date was chosen because approximately 60% of all transactions occur before this date and 40% occur after. We use a temporal split rather than a user-ID split to prevent leakage of future market information into past behavior. In financial time series, user behavior often reflects contemporaneous market conditions, so splitting by calendar time ensures that the training data precede the test data chronologically. In addition to the cutoff, we included a buffer window at the end of both the training and testing sets, during which time we ignored new index events. We did this to give each index event a fair amount of time to see an outcome

Table 2: The structure of one of our survival datasets, Borrow-to-Repay, with some features excluded for brevity. "Time" represents the elapsed number of seconds between the index and outcome events. "Status" represents whether the observation was censored. "User" is a blockchain address of the user who performed the transaction. "Coin" is the cryptocurrency used in transactions. "Index" and "Outcome" are the index and outcome event types.

| Time | Status | User | Coin | Index | Outcome | Amount ($) | $\cdots$ |
|---|---|---|---|---|---|---|---|
| 15,487 | 1 | 0xab123... | DAI | Borrow | Repay | 15,000.00 | $\cdots$ |
| 190,601 | 1 | 0x98si4... | USDT | Borrow | Repay | 1,349.97 | $\cdots$ |
| $\vdots$ | $\vdots$ | $\vdots$ | $\vdots$ | $\vdots$ | $\vdots$ | $\vdots$ | $\ddots$ |
| 173,472 | 0 | 0x74flk... | USDC | Borrow | Repay | 598.80 | $\cdots$ |

Table 3: Overview of all the survival data in our suite of 16 datasets.

| Attribute | Description |
|---|---|
| Total # Records | 7,698,497 |
| Time Period | November 30, 2020 - September 30, 2024 |
| Subjects | Users, Coins |
| Unique Users | 114,861 |
| Unique Coins | 60 |
| Index Events | Borrow, Deposit, Repay, Withdraw |
| Outcome Events | Account Liquidated, Borrow, Deposit, Repay, Withdraw |
| Mean Censoring Rate | 81.26% |
| # Features | 128 |
| Data Source | The Graph |

event before the end of the observation period. We included a 30-day buffer at the end of both the training and testing windows, during which new index events are ignored. This reduces edge effects and ensures index events near the boundary have adequate follow-up time for outcomes to occur, improving the validity of time-to-event evaluation on the test set.

## 3 FinSurvival Prediction Tasks and Results

### 3.1 Model Selection and Evaluation Metrics

We implemented six models for survival prediction: two linear models, two tree-based ensemble models, and two deep-learning models. The linear models we implemented were Cox Proportional Hazards [Cox (1972)] and Accelerated Failure Time (AFT) [Wei (1992); Crowther et al. (2022)]. The Cox model estimates hazard rates as a function of user and transaction features, while the AFT model predicts the log-transformed survival time under a Weibull distribution assumption.

Table 4: Summary statistics of the different types of survival data in the dataset based on index and outcome events.

| Index Event | Outcome Event | FinSurvival Stats | | |
| --- | --- | --- | --- | --- |
| | | # Records | Mean   Delay | Censored % |
| borrow | liquidated | 264,536 | 323.30 | 83.42 |
| borrow | deposit | 640,497 | 246.85 | 87.50 |
| borrow | repay | 267,010 | 78.80 | 16.62 |
| borrow | withdraw | 578,605 | 274.43 | 89.50 |
| deposit | liquidated | 507,437 | 375.71 | 98.73 |
| deposit | borrow | 644,296 | 303.62 | 87.83 |
| deposit | repay | 595,761 | 313.39 | 86.43 |
| deposit | withdraw | 629,082 | 198.42 | 57.15 |
| repay | liquidated | 208,939 | 333.97 | 93.18 |
| repay | borrow | 226,764 | 192.42 | 57.65 |
| repay | deposit | 573,804 | 242.44 | 91.25 |
| repay | withdraw | 514,555 | 261.04 | 90.65 |
| withdraw | liquidated | 399,704 | 360.26 | 99.34 |
| withdraw | borrow | 555,420 | 298.85 | 93.26 |
| withdraw | deposit | 587,063 | 236.85 | 76.73 |
| withdraw | repay | 505,024 | 307.11 | 90.98 |

The two tree-based ensemble models were Random Survival Forests (RSF) [Ishwaran et al. (2008); Ishwaran and Kogalur (2007)] and XGBoost [Chen and Guestrin (2016)]. RSF extends the random forest framework to time-to-event data by constructing an ensemble of survival trees, each fit on a bootstrap sample of the data and using log-rank splitting to maximize separation of survival times. Predictions are obtained by aggregating survival functions across trees, enabling the model to capture complex nonlinear effects and feature interactions while handling censoring naturally. XGBoost applies a gradient-boosted decision tree framework to survival analysis. In our implementation, it uses an AFT objective with a logistic loss distribution to directly model censored event times.

The two deep-learning survival models we implemented were DeepSurv [Katzman et al. (2018)] and DeepHit [Lee et al. (2018)]. Over the past decade, numerous deep survival models have been introduced, leveraging state-of-the-art deep learning architectures such as feed-forward neural networks and transformer Models [Wiegrebe et al. (2024)]. Notably, there are Cox-based models like DeepSurv and discrete-time methods like DeepHit. DeepSurv uses a feed-forward neural network to model the log-risk function within a traditional Cox regression framework. DeepHit models survival in discrete time by learning a probability mass function over time intervals (single-risk in our case), yielding discrete hazard or event-time probabilities that are trained with a survival-appropriate loss. We implemented these two models using previously published code and compared their results with the traditional models on our data.

Recent years have seen a rapid expansion of deep survival architectures beyond these two approaches. Among Cox-based extensions, methods such as Cox-Time [Kvamme et al.

(2019b)] relax the proportional-hazards assumption by allowing time-varying effects, while NN-DeepSurv [Tong and Zhao (2022)] integrates regularization for handling missing features. Other variants such as Cox-nnet [Ching et al. (2018)] focus on modeling high-dimensional omics or multimodal data, and some like CNN-Cox [Yin et al. (2022)] adapt more complex architectures, including using convolutional neural nets for unstructured imaging input.

On the discrete-time side, extensions of DeepHit have explored temporal and multimodal structure in censored data. Dynamic-DeepHit'[Lee et al. (2019)] employs recurrent neural networks to model longitudinal covariates, while TransformerJM [Lin and Luo (2022)], among others, introduce transformer-based architectures that jointly learn from survival and time-series information through combinations of likelihood, ranking, and reconstruction-based losses. These approaches demonstrate the growing diversity of survival-specific deep architectures that incorporate temporal dynamics, attention, and multimodal fusion.

In our benchmark, we focus on widely adopted representatives of DeepSurv and DeepHit which represent both a continuous-time and a discrete-time paradigm, respectively. Both are foundational baselines against which newer deep-learning models of all kinds can be evaluated on large-scale, structured financial survival data such as FinSurvival.

Detailed descriptions of each model's implementation, including data preparation, feature selection, and hyperparameter settings, are provided in Appendix A.

The Concordance Index (C-index) evaluated the baseline survival prediction models. The C-index assesses the models' abilities to discriminate between individuals with different survival times. C-index values are better the closer they are to 1, and the lowest possible C-index is 0.5. We primarily use the Concordance Index throughout our experiments as implemented in the `survival` package [Therneau (2022)]. Harrell's C-index measures the fraction of all comparable subject pairs for which predicted and observed event times are concordant. It is the standard metric for continuous time-to-event outcomes and is appropriate for both parametric and non-parametric models. We use Harrell's C-index for the five continuous-time models we implemented (Cox, AFT, RSF, XGBoost, and DeepSurv). For DeepHit, which models survival in discrete time by predicting a probability mass function over event intervals, we instead report Antolini's C-index (Antolini et al., 2005) as implemented in `pycox`. Antolini's formulation generalizes Harrell's C-index to discrete-time settings by comparing predicted survival probabilities at each observed event time, properly accounting for ties and censoring. Both indices are consistent measures of discriminative performance, and higher values indicate stronger concordance. We emphasize that our experimental protocol was finalized before running any models; the benchmark design and evaluation metric were pre-specified to avoid "C-hacking".

To assess which models performed best overall, we computed the mean Borda rank for each model as described in Pavão (2023). First, we ranked the models on each dataset using the Borda ranking system, which assigns each candidate model a rank from 1 to $n$, where $n$ is the number of candidates, based on their score. For example, since we have $n = 6$ models, for any one dataset, the model with the highest C-index is given a rank of 1, the second highest gets rank 2, etc. With models being ranked for each dataset, we computed the mean Borda rank for each model to estimate which models were the best across the whole benchmark. We sort the columns according to this ranking. Lower Borda ranks mean better performance.

We use the same method to rank the difficulty of the datasets. By ranking how well individual models performed across all datasets and subsequently averaging these rankings, we can compare the difficulty of each dataset for our models to learn. We sort the rows in order of increasing mean Borda rank to represent the increasing difficulty of the datasets. Since both the rows and columns are sorted by their mean Borda rank, which is dependent on the number of rows or columns, we divide the mean Borda ranks by the number of rows and columns, respectively, to have the same scale on each axis.

## 3.2 FinSurvival Prediction Benchmark Results

In Figure 3, we summarize the performance of the six survival models on our datasets (Table 6 in Appendix A.5 gives the exact values). The rows and columns of the heatmap are sorted based on their mean Borda ranks.

Overall, model performance is consistently high, with C-index values generally ranging between 0.75 and 0.85 across datasets. Among the models, RSF achieved the highest average C-index and the best overall mean Borda rank, indicating that ensemble-based methods remain particularly effective for large-scale, high-dimensional financial survival data. DeepSurv ranked second overall and performed comparably to RSF on many datasets, highlighting the strength of neural representations for modeling nonlinear interactions in censored time-to-event data. XGBoost, trained with an AFT objective, followed closely behind, demonstrating that gradient-boosted tree ensembles remain a strong baseline for structured survival prediction.

DeepHit also performed competitively, ranking ahead of the linear models and achieving stable results across datasets. The Cox and AFT models produced consistent but lower C-indices, reflecting the limitations of linear assumptions in capturing the complex temporal and behavioral dynamics present in this data.

With the rows and columns arranged by mean Borda rankings, the heatmap shows the best-performing models on the left and the easiest-to-learn datasets on top. The models perform worse, and the datasets get harder as we move towards the bottom right. There are some interesting things to learn from this ranking. Across datasets, a clear gradient of difficulty emerges. Tasks with withdrawals as index events tend to be the easiest to predict, while tasks with borrows as index events tend to remain more challenging. Withdraws as index events being easy to predict feels a little bit counterintuitive since withdrawals involve pulling money out of the lending pool, and it does not feel like they set up an obvious next action for a user. Additionally, as an outcome event, account-liquidated consistently makes for difficult datasets to predict. Also, two of the more obviously meaningful datasets, deposit-withdraw and borrow-repay, are among the more difficult datasets to predict. These events could easily be interpreted as representing the time until a customer churns and the time until a loan repayment is made, respectively. That they remain difficult to predict shows that user behavior is varied and based on a complex set of factors.

## 4 Discussion

The FinSurvival benchmark highlights clear distinctions among modeling paradigms when applied to large, high-censoring financial survival data. Tree-based ensembles like RSF and

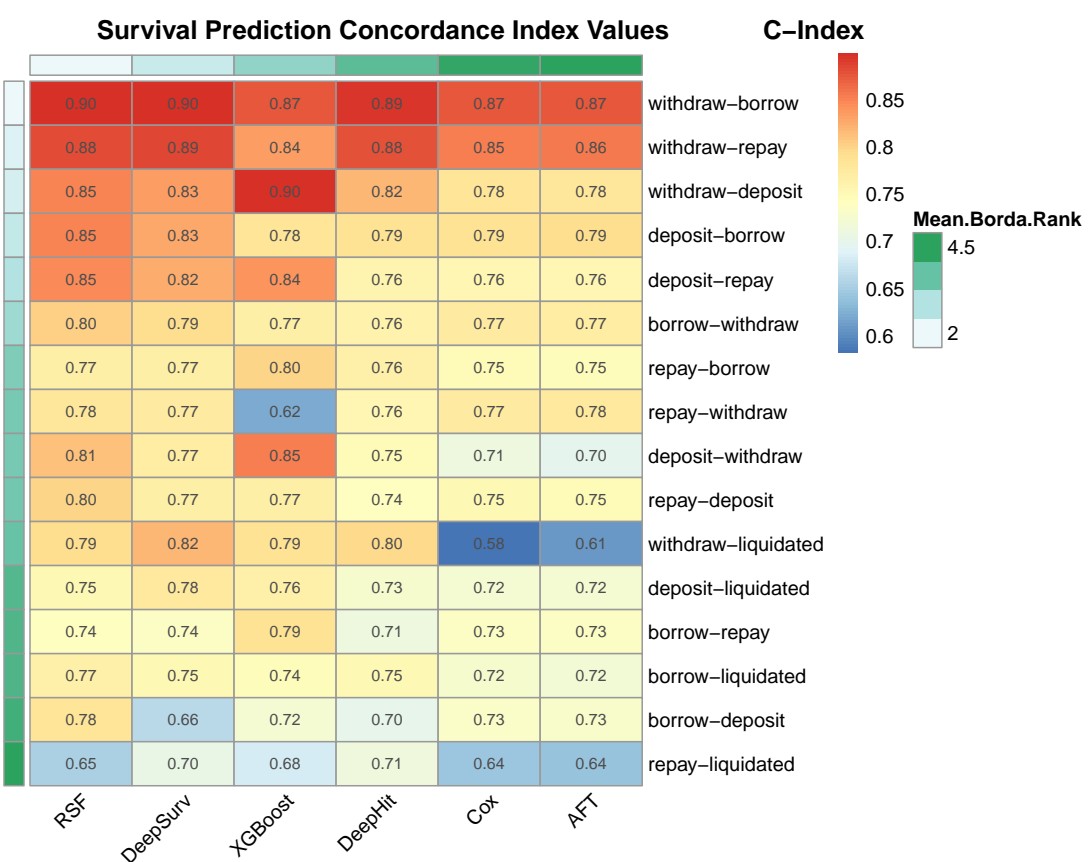

Figure 3: Heatmap displaying prediction C-index values for survival outcomes across different index-outcome event pairs. Each cell represents one model's C-index score for an individual dataset. The rows and columns are each ordered in decreasing order based on the mean Borda rank among the rows and columns, respectively. Models on the left side performed better on average, and datasets towards the top of the heatmap were generally easier for models to learn.

XGBoost achieve the strongest discrimination overall, demonstrating their ability to model complex feature interactions and mixed data types without heavy parametric assumptions.

At the same time, deep survival models like DeepSurv and DeepHit show strong and reliable performance, often rivaling tree-based ensemble approaches. These results indicate that neural architectures can effectively learn nonlinear temporal representations from transactional survival data when appropriately tuned and evaluated.

Linear models such as Cox and AFT continue to serve as valuable baselines due to their interpretability, computational efficiency, and well-established theoretical grounding. However, they struggle to match the flexibility of nonlinear approaches on high-dimensional, behaviorally driven data.

Regarding the effect of censoring rates on model performance, Table 4 shows that censoring varies widely across tasks, from 16.6% for borrow-to-repay up to approximately 99% for several of the liquidation outcomes. However, examining these censoring rates in combination with the concordance index values across tasks indicates that the censoring rate does not determine predictive difficulty. For example, borrow-to-repay has the lowest censoring rate (16.6%) yet on average across all models has one of the lower C-index values ($\approx 0.74$); meanwhile, the withdraw-to-borrow task has very high censoring (93.3%) but is the easiest task on average across the models with a mean C-index of $\approx 0.89$. Across all 16 tasks, the Pearson correlation between censoring rate and mean C-index is very small ($r < 0.1$), indicating that censoring is not a primary driver of performance differences. The tasks with liquidation as the outcome show how heavy censoring can coincide with lower performance. For example, the repay-to-liquidated task has the lowest average performance with a mean C-index of 0.67, and a high censoring rate at 93.2% censored. This suggests that these outcomes are both rare and harder to rank from features. Overall, this benchmark suggests that task difficulty is primarily determined by the underlying behavior pattern (index-outcome combination) and the strength of feature signal conditional on observing an event.

Our results demonstrate that the FinSurvival dataset is demanding and complex, offering a rigorous benchmark for evaluating and advancing AI survival models. This work highlights opportunities for improved deep learning strategies and hybrid approaches to better capture the intricate patterns inherent in large-scale financial survival data. Overall, FinSurvival establishes a challenging new testbed for survival modeling, motivating further research into deep and hybrid methods that better capture the behavioral complexity of financial survival data.

## Broader Impact Statement

Our FinSurvival benchmark and datasets fill a critical need for large-scale survival datasets in finance. Unlike traditional survival data limited to small samples or sensitive domains like medical records, our openly accessible DeFi-based transaction data offers millions of records covering loans, deposits, and liquidations. By capturing high censoring rates and complex user behaviors at scale, FinSurvival drives methodological innovation in deep learning for survival modeling. It can also be helpful for quantifying things like credit risks, repayment patterns, etc., in a transparent manner. The openness and scale of this benchmark fill in

an important gap that should help improve the accuracy and modeling capabilities of AI methods for survival analysis and help extend survival research to new domains. While there are potential ethical, privacy, and fairness concerns associated with the release and use of traditional financial data, these are largely ameliorated in these DeFi sets. DeFi users inherently consent to make their account transactions public by engaging with a DeFi protocol on a public blockchain. There is no personal or private information associated with account IDs. Also, DeFi protocols such as Aave are inherently fair because every action of the protocol is encoded in published smart contracts that apply identically to all accounts as described by public data on the blockchain. This contrasts with the well-documented biases that have been found in traditional lending, e.g., red-lining in mortgage lending Yinger (2018). The possibility that the survival models produced could result in unethical and fraudulent behavior is very small.

## 5 Conclusion and Future Work

This suite of survival datasets represents only a small fraction of the transaction data available through DeFi protocols. Not only are there numerous other deployments of the Aave protocol on other blockchains that have far more transaction data than the Ethereum chain used for these datasets, but there are other lending protocols, and other DeFi protocol types like Decentralized Exchanges (DEXs) that generate huge amounts of transaction data for unique problems. Our data creation pipeline can easily be used to convert transaction data from all of these protocols into new, massive survival data to create additional use cases and further challenges for survival modeling.

One way these datasets could be expanded is by adding more features. An interesting avenue of expansion could be incorporating exogenous data such as stock prices and cryptocurrency prices over time, along with other data typically associated with price data, like trade volume, volatility, etc. These could be joined with our existing data based on the timestamp and may help models have more predictive power.

The features created for this work were hand-crafted based on domain knowledge and intuition as to what features might be useful for prediction. This process takes a long time and extensive domain knowledge. One way this data could be used better is through an automatic feature-creation process that leverages AI techniques such as BERT-style transformer-based models to auto-generate more interesting features based on the raw transaction data. Such a model could create embeddings for each index event that encode a user's entire transaction history and capture more mathematically complex relationships in the data.

One limitation of this work, as it currently stands, is that it does not take into account competing risks. Different outcome events can "compete" with one another as possible outcomes for the same index event. For instance, if a user's borrowed funds are liquidated, this competes with their ability to repay that loan, i.e., borrow. Our survival modeling pipeline can include competing events in creating datasets, but we have not included them in this analysis [Green et al. (2024)]. Incorporating competing events in these analyses would help the models more accurately reflect the dynamics of user behavior within the underlying transaction data.

## Acknowledgments and Disclosure of Funding

We would like to thank students in the Data Analytics Research Course at Rensselaer who contributed to the creation of this project: Abid Talukder, Jianzhuo Liu, Kaiyang Chang, Sean Fitch, Haolin Luo, Charlotte Newman, Emmet Whitehead, Daniel Kirtman, Campbell Drahus, Alejandro Laphond, and David Quintero. The authors acknowledge the support from NSF IUCRC CRAFT center research grants (CRAFT Grant #22015) for this research. The opinions expressed in this publication and its accompanying code base do not necessarily represent the views of NSF IUCRC CRAFT.

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

## Appendix A. Model Implementation Details

Each of the four non-deep-learning methods were implemented in R using publicly-available packages. The two deep-learning models were implemented in Python.

### A.1 Linear Models

**Preprocessing**   Both of the linear models, Cox and AFT, follow this preprocessing pipeline: (1) scaling the data, (2) encoding categorical data, and (3) applying Principal Component Analysis (PCA) to reduce the size of the data.

For scaling the data, we mean-center the training data using R's built-in `scale` function. We use the training mean and standard deviation to scale the testing data as well. This scaling is only applied to numeric features.

We do two things to handle categorical data. First, for any categorical feature with more than ten categories, we keep the ten most frequent categories from the training set and combine the remaining into another category called "Other". We keep the same ten categories in the testing set and make "Other" for any remaining data as well. We do this so models have an easier time learning about less-common categories and to ensure the categories seen in the testing set are the same as in the training set. After this, we apply one-hot encoding to the categorical features using the `fastDummies` package in R [Kaplan (2025)].

Finally, we apply PCA to the resulting data to reduce collinearity among the features. We use the `prcomp` function from `stats` package [R Core Team (2024)]. We compute the principal components on the training set and keep the principal components that explain 90% of the variance in the data. We use the same set of principal components on the testing set.

**Cox Proportional Hazards Model:**   We trained the Cox model using the `coxph()` function from the survival package [Therneau (2022)] with its default settings. The model estimates a log-risk (hazard) score as a linear combination of the covariates, corresponding to the proportional hazards assumption. Predictions were obtained using `type='lp'` to return the linear predictor (log-risk) values, which preserve the relative risk ranking across samples and are therefore appropriate for C-index evaluation. No additional hyperparameters were tuned beyond the default convergence settings, as the Cox model's partial-likelihood formulation is convex and deterministic given the data and features.

**Accelerated Failure Time Model:**   Accelerated Failure Time models [Crowther et al. (2022)] are an alternative to the commonly used proportional hazards model. In an AFT model, the effect of the covariates accelerates or decelerates the survival time by a specific factor. This acceleration factor is assumed to be constant. Our implementation of the AFT model uses the `survreg` function from the `survival` package, and we use the Weibull distribution to fit the data. Any other parameters use the default values.

### A.2 Tree-based Models

**Preprocessing**   Tree-based models use the shared base feature representations but to not apply scaling, one-hot encoding, nor dimensionality reduction via PCA in the preprocessing phase. This is because these models are invariant to feature scaling, can directly exploit discrete variables, and perform implicit feature selection.

**Random Survival Forest (RSF)**   We trained RSF using the `rfsrc.fast()` function from the randomForestSRC package [Ishwaran and Kogalur (2024)]. Random Survival Forests extend the standard random forest framework to time-to-event data by growing an

ensemble of survival trees using log-rank splitting criteria to maximize separation among survival times while naturally handling censoring. The `rfsrc.fast` function approximates random forests using subsampling in order to speed up the computation. All model hyperparameters were left at their default values according to the package documentation.

**XGBoost**   Extreme Gradient Boosting (XGBoost [Chen and Guestrin (2016)]) is a distributed gradient-boosting decision tree algorithm. We train the model using the `xgb.train` function from the `xgboost` package [Chen et al. (2024)]. The model was trained with an AFT objective. We use a maximum of 1,000 boosting iterations and use a validation set to check the loss every 50 iterations, stopping early if the validation loss does not improve.

### A.3 Deep Learning Models

Both deep learning models (DeepSurv and DeepHit) were implemented in Python using the `pycox` library [Kvamme et al. (2019a)]. These frameworks enable the construction of fully connected multilayer perceptrons (MLPs) that learn nonlinear representations of censored time-to-event data through appropriate survival loss functions.

**Preprocessing**   Before training, we applied the following preprocessing pipeline:

1. **Feature cleaning and encoding.** Identifier columns (`id`, `user`, `pool`, `Index Event`, `Outcome Event`, `type`, `timestamp`) were removed. Categorical features were detected from the training data, capped to their ten most frequent categories, and one-hot encoded using the training set as the column template.

2. **Standardization and variance filtering.** All numeric features were standardized using a `StandardScaler` fitted on the training data. Features with near-zero variance ($\leq 10^{-5}$) were dropped.

3. **Feature selection.** To limit dimensionality, we applied Minimum Redundancy–Maximum Relevance (MRMR) selection on the training set, retaining the top features most associated with the event indicator.

4. **Downsampling censored observations.** If uncensored observations accounted for less than 5.5% of the data, we retained all events and randomly subsampled censored rows to reach this ratio.

5. **Temporal filtering.** Records with survival times of 0 (i.e. the label `timeDiff` $= 0$) were removed, and survival durations were converted from seconds to days.

**Validation set construction**   All transformations were fit exclusively on the training data and applied to validation and test sets without refitting, ensuring no data leakage. The validation sets were constructed exclusively from the training data. After all preprocessing was applied to the training data, the resulting training set was randomly divided into training and validation subsets using an 80/20 split. This split was stratified by the event indicator to preserve the proportion of censored and uncensored observations. The validation set was used only for hyperparameter selection and early stopping during model training. The test sets were never used during tuning.

### A.3.1 DEEPSURV

DeepSurv (Katzman et al., 2018) extends the Cox proportional hazards model by parameterizing the log-risk function with a neural network. The network produces a scalar risk score proportional to the log-hazard and is trained to minimize the negative log partial-likelihood of the Cox model.

**Implementation.** We implemented DeepSurv using `CoxPH` from `pycox` with a custom `MLPVanilla` network. Batch normalization was enabled between layers. The model was trained using mini-batches and early stopping on an internal validation set.

**Hyperparameter tuning.** A random-search procedure explored the hyperparameter ranges seen in Table 5. Deeper networks used smaller learning rates and trained for approximately 10 epochs; shallower networks used larger learning rates and approximately 5 epochs. Gradient clipping was conditionally applied based on network depth, activation type, and optimizer choice to prevent exploding gradients. The final model outputs a log-hazard risk score, which was negated prior to computing Harrell's C-index using `lifelines.utils.concordance_index`. 30 hyperparameter configurations were tried for each task.

Table 5: The hyperparameters and associated search spaces used for both DeepHit and DeepSurv during training.

| Hyperparameter | Range / Options |
| --- | --- |
| Number of layers | 2–18 |
| Hidden layer size | {16, 32, 64, 128, 256, 512} |
| Dropout rate | 0.0–0.5 |
| Batch size | {256, 512, 1024, 2048, 4096} |
| Learning rate | {1e-4, 5e-4, 1e-3, 1e-5, 5e-5} (depth-dependent) |
| Weight decay | {0, 1e-5, 1e-4, 1e-3} |
| Optimizer | Adam, AdamW, RMSprop |
| Activation | ReLU, LeakyReLU, ELU, SiLU, GELU, Tanh |
| Batch normalization | Enabled |

### A.3.2 DEEPHIT

DeepHit (Lee et al., 2018) models the conditional event-time distribution in discrete time by predicting a probability mass function over $N$ time intervals. It jointly optimizes a weighted sum of a negative log-likelihood and a ranking loss to balance calibration and concordance.

**Implementation.** We used the `pycox.models.DeepHitSingle` class with the same MLP backbone as DeepSurv and automatic discretization of event times into 100 intervals. The model learns a discrete survival probability at each interval, from which cumulative survival curves can be computed.

**Hyperparameter tuning.** The same random-search strategy was applied, sharing the parameter ranges for architecture, optimizer, and regularization found in Table 5. The loss hyperparameters followed the defaults $\alpha = 0.2$ and $\sigma = 0.1$. Learning rates and epochs followed the same depth-dependent schedule, with optional gradient clipping applied

according to the heuristic rules described for DeepSurv. 30 hyperparameter configurations were tried for each of the sixteen tasks.

**Evaluation.** Predicted survival functions were evaluated using the Antolini C-index, computed with `pycox.evaluation.EvalSurv`, which properly accounts for censoring and tied events in discrete time.

## A.4 Evaluation Metrics

Model performance was primarily evaluated using Concordance Indices that measure how well each model discriminates between subjects with different observed event times. For continuous-time models (Cox, AFT, RSF, XGBoost, and DeepSurv), we use Harrell's C-index [Harrell et al. (1982)], implemented via the `survival` package [Therneau (2022)]. This metric computes the fraction of all comparable subject pairs for which the predicted and observed orderings of event times agree. Because some models predict hazards (where larger values correspond to shorter survival) while others predict survival times, we use the `reverse=TRUE` option for hazard-based models to ensure consistent directionality across methods.

For DeepHit, which produces discrete-time survival probabilities rather than continuous risk scores, we use Antolini's C-index [Antolini et al. (2005)], computed using the `pycox.evaluation.EvalSurv` module. Antolini's C-index extends Harrell's approach to discrete survival predictions by comparing predicted survival probabilities at each event time while correctly handling ties and censoring. Thus, while Harrell's C-index is used for all continuous-time models, Antolini's C-index is the appropriate analog for DeepHit's discrete-time formulation.

## A.5 Model Results

Table 6 gives the full numeric results for all models across all datasets.

## Appendix B. Survival Data Creation Pipeline

### B.1 Raw Transaction Data Collection

The first phase for creating this dataset was collecting the raw market transaction data from the Aave platform. The underlying transaction data upon which this data is based comes from The Graph[6], a decentralized protocol for indexing and querying data from blockchains that primarily target the Ethereum network. Specifically, we collect transaction data from the Aave V2 subgraph[7]. This subgraph contains many tables necessary to query to get a comprehensive view of the transactions in Aave.

The data for each transaction type is in its table, so we queried each type to get data for all transactions from November 30, 2020, through September 30, 2024. The transaction types we use in our dataset are Deposits, Withdrawals, Loans, Repayments, and Liquidations.

---

6. thegraph.com

7. https://thegraph.com/hosted-service/subgraph/aave/protocol-v2?version=current

Table 6: Performance evaluation of survival regression models on various datasets using the Concordance Index. Best results per row are in bold.

| Dataset | XGBoost | AFT | DeepHit | DeepSurv | Cox | RSF |
|---|---|---|---|---|---|---|
| borrow-repay | **0.788** | 0.734 | 0.709 | 0.737 | 0.734 | 0.738 |
| borrow-deposit | 0.723 | 0.730 | 0.700 | 0.662 | 0.732 | **0.784** |
| borrow-withdraw | 0.769 | 0.772 | 0.762 | 0.792 | 0.772 | **0.803** |
| borrow-liquidated | 0.745 | 0.721 | 0.750 | 0.751 | 0.724 | **0.766** |
| repay-borrow | **0.798** | 0.745 | 0.764 | 0.766 | 0.749 | 0.766 |
| repay-deposit | 0.768 | 0.749 | 0.739 | 0.768 | 0.751 | **0.798** |
| repay-withdraw | 0.620 | 0.778 | 0.755 | 0.774 | 0.773 | **0.782** |
| repay-liquidated | 0.681 | 0.640 | **0.712** | 0.705 | 0.643 | 0.652 |
| deposit-borrow | 0.782 | 0.793 | 0.787 | 0.829 | 0.786 | **0.849** |
| deposit-repay | 0.840 | 0.756 | 0.758 | 0.825 | 0.756 | **0.848** |
| deposit-withdraw | **0.854** | 0.698 | 0.748 | 0.771 | 0.712 | 0.813 |
| deposit-liquidated | 0.763 | 0.722 | 0.726 | **0.777** | 0.721 | 0.751 |
| withdraw-borrow | 0.874 | 0.874 | 0.894 | **0.898** | 0.874 | 0.897 |
| withdraw-repay | 0.835 | 0.856 | 0.880 | **0.885** | 0.855 | 0.883 |
| withdraw-deposit | **0.899** | 0.779 | 0.818 | 0.835 | 0.784 | 0.849 |
| withdraw-liquidated | 0.788 | 0.608 | 0.796 | 0.817 | 0.581 | **0.793** |
| Mean C-Index | 0.783 | 0.747 | 0.769 | 0.787 | 0.747 | **0.798** |
| Mean Borda Rank | 3.250 | 4.812 | 4.000 | 2.438 | 4.688 | **1.812** |

To get all pertinent information for each transaction, we also collect data from the ReserveParamsHistoryItems from the same time span. It provides timestamped information

about reserves whenever they are used in a transaction within Aave. This is how we can get information about how much a reserve was worth and what its interest rates were at the time of each transaction.

Finally, we also collected data about the individual coins (the table is called Reserves) to get basic information about each reserve, such as its symbol, what functionality is available for it in Aave, and how many decimal places to adjust its numerical values by. With the information from all of these tables, we were able to create one unified and human-readable view of the transaction data. We combined all of the transaction-type-specific tables, sorted them chronologically by their UNIX timestamp, and replaced IDs in many columns with more pertinent information so that each transaction could be comprehensible to a human. Information about the coins used in each transaction was added explicitly to each transaction record, so that at each transaction we can see the symbol of the coin(s) involved (e.g., BTC, ETH). The amounts of each currency being used in each transaction were adjusted by the currencies' specific decimal exponent, as well as their conversion factors to USD at each transaction time. Other time-dependent, currency-specific data was added to each transaction to increase the amount of information contained within one record.

The final structure of this transaction-level data is shown in Table 7. This table does not include all the columns of the data, as there are too many features to include and many of them are transaction-type-specific. We also provide a table of metadata about these transactions in Table 8.

Table 7: Structure of raw transaction data showcasing the main features present for all transaction types.

| Datetime | Type | User | Coin | Amount | Amount ($) | $\cdots$ |
|----------|------|------|------|--------|------------|----------|
| 11-30-2020 23:15:00 | Deposit | <ID> | USDT | 100.00 | 100.00 | $\cdots$ |
| 11-30-2020 23:15:30 | Borrow | <ID> | XSUSHI | 15.52 | 100.00 | $\cdots$ |
| $\vdots$ | $\vdots$ | $\vdots$ | $\vdots$ | $\vdots$ | $\vdots$ | $\ddots$ |
| 12-31-2023 23:50:00 | Repay | <ID> | DAI | 25,000.667 | 24,978.34 | $\cdots$ |
| 12-31-2023 23:50:45 | Withdraw | <ID> | WETH | 3.652 | 8,976.09 | $\cdots$ |

## B.2 Transformation to Survival Data

Phase two of creating this data involved the creation of the pipeline to convert transaction data into survival data. Survival data, also known as time-to-event data, involves observations where the outcome of interest is the time until a specific event occurs. This type of data is characterized by two main components: the observed time $T_i$, which is either the time until the event occurs or the time until the last follow-up (for censored data), and the event indicator $\delta_i$, which denotes whether the event has occurred ($\delta_i = 1$) or the observation

Table 8: Metadata about the transaction data that we collected, cleaned, and used to create the survival data being published with this paper.

| Attribute | Description |
| --- | --- |
| Total Transactions | 1,977,491 |
| Total Users | 117,008 |
| Number of Features | 38 |
| Time Span | November 30, 2020 - September 30, 2024 |
| Size of CSV | 668.8 MB |
| Data Source | The Graph |

is censored ($\delta_i = 0$). Additionally, survival data often includes a set of covariates $\mathbf{X}_i$ that represent other variables which might influence the time-to-event. Mathematically, a survival dataset for $n$ subjects is represented as $\{(T_i, \delta_i, \mathbf{X}_i) : i = 1, 2, \ldots, n\}$. This structure enables the analysis of both the timing of events and the factors that affect these timings.

The motivating idea for how our DeFi transaction data can become survival data is that each transaction a user performs could be considered an "index event," triggering the start of a record which lasts until a future transaction of interest, which could be considered the "outcome event." For instance, if we want to track how long it takes for a user to pay off a loan of a certain currency, we could treat the transaction where a user borrows that currency as the index event and track that user until they make a repayment of that same currency. With this idea in mind, our pipeline for creating survival data needs the following parameters:

- **Event Data:** A tabular dataset containing all recorded events that could be relevant to creating the survival dataset.

- **Subjects:** A specified set of one or more columns of the event data that define who/what the subjects will be that we track when creating the survival data. For instance, most of the time we want to see both the "user" and the "coin" columns, because we are interested in how an individual user interacts with a specific coin over time.

- **Observation Period:** A start and end date and time over which to compute the survival data. By default, this can be the entire duration of the events dataset, but could be set to a shorter time window for more targeted analysis.

- **Index Event Set:** One or more event types which will be treated as the index events to start the tracking of survival records.

- **Outcome Event Set:** One or more event types that will trigger the end of a survival record if it occurs following an index event made by the same subject(s).

With these parameters defined, we create the pipeline for survival data creation. Given all the parameters, we first filter out all events that do not occur within the specified observation period. Then we create a subset of the data that only includes the desired

index events and another subset that only includes the desired outcome events. Grouping these subsets by the selected subjects, we then perform a rolling join on the index events with the outcome events, matching the subjects and using the first event after each index event. This creates a table where each row has information on an index event and the first outcome event performed by the same subject, if any, performed after the index event. With this, we can calculate the elapsed time between the two events, using the final time of the observation period as the outcome event time if no appropriate outcome event occurred.

## B.3 Curating Survival Datasets

Considering each of the five transaction types as possibilities for index and outcome events was the obvious way to go about this, but liquidation events need to be handled more carefully than the others due to their involving multiple parties. So, first we considered the "basic" transactions of borrows, repays, deposits, and withdraws. If we choose one of these transaction types as an index event (e.g., borrows) and another type as outcome events (e.g., repays), we can create survival data that answers a question like "How long do users take to repay after borrowing?" It is important to note that the choice of subjects here is both the user and the currency involved in the transaction, because if an index event shows a user borrowing e.g. Wrapped Bitcoin, an appropriate outcome event should be that same user repaying Bitcoin, not a different currency for which they might also have a loan. Putting these ideas together, we created 12/16 datasets using the different combinations of basic transactions as the index and outcome events.

We wanted to include survival datasets using liquidations. Liquidations can occur when a user's overall account in Aave has an "unhealthy" balance of deposited assets compared to borrowed assets. These assets can include a variety of different currencies. When a user's account is unhealthy, another user can perform a liquidation transaction, paying off a portion of the unhealthy user's loans to claim an equivalent portion of the user's deposited collateral assets and a small liquidation bonus from the protocol as an incentive. So, liquidation transactions include more than just one user. They include a "liquidator", which is the user who performs the liquidation transaction, and a "liquidatee" whose account is being liquidated. Additionally, a liquidation transaction can involve any one of a user's borrowed currencies as the "principal" currency, and any one of the user's deposited currencies as the "collateral" currency. Given all of this, we handle liquidations differently than the other transaction types. In our suite of datasets, we only consider the case when a user's account is liquidated. We use this event exclusively as an outcome event.

## B.4 Feature Construction

The feature engineering process produced a total of 128 features derived from raw transaction data on AAVE Mainnet V2, of which 106 are constructed from the base features. These features fall into four primary categories: base features, user history features, market history features, and time features. The base features refer to the original fields extracted from the raw data, such as transaction amounts, timestamps, and coin types. From these base features, additional derived features were created to capture more complex relationships and non-linear patterns relevant to survival prediction up to and including the index event. The full list of base features can be found in Table 9.

Table 9: The 24 features from the raw transaction data and the transaction types for which each feature is relevant.

| Feature Name | Relevant Transaction Types |
| --- | --- |
| timestamp | All |
| user | All |
| pool | All |
| type | All |
| reserve | Borrow, Deposit, Repay, Withdraw |
| coinType | Borrow, Deposit, Repay, Withdraw |
| amount | Borrow, Deposit, Repay, Withdraw |
| amountUSD | Borrow, Deposit, Repay, Withdraw |
| amountETH | Borrow, Deposit, Repay, Withdraw |
| borrowRate | Borrow |
| borrowRateMode | Borrow |
| liquidator | Liquidation |
| principalAmount | Liquidation |
| principalReserve | Liquidation |
| principalReserveType | Liquidation |
| principalAmountUSD | Liquidation |
| principalAmountETH | Liquidation |
| collateralAmount | Liquidation |
| collateralReserve | Liquidation |
| collateralReserveType | Liquidation |
| collateralAmountUSD | Liquidation |
| collateralAmountETH | Liquidation |
| priceInUsd | Borrow, Deposit, Repay, Withdraw |
| version | All |
| deployment | All |

The user history features are features created by user. For each user, cumulative calculations of the following quantities are created: seconds since first transaction, seconds since last transaction, the count of each transaction types a user has made, the sum and average amount of each transaction type a user has made (in the native amount, Dollars, and Ethereum). The full list of these features can be found in Table 10.

Similarly to user history features, market history features compute metrics for the entire Aave V2 Mainnet market. This is a useful way of approximating a market's relative supply and demand at a given point. For the entire blockchain, features track the number of each type of transaction, the average amount for each type of transaction, and the average and sum amounts of all currencies together in dollars and Ethereum. The full list of these features can be found in Table 11.

Time features were computed for every observation in the raw data. The 'timestamp' variable represents the POSIX time (time since January 1, 1970, 00:00:00 UTC) in seconds. Circular representations of these time features were also created to capture the cyclical

nature of time, such as the closeness of 11:59 PM and 12:00 AM. This resulted in three additional features per time interval: the value, sine, and cosine. The full list of time features can be found in Table 12.

The data set created contains 128 features in total, of which 106 are constructed from the base features.

## B.5 Feature Evaluation and Extension

Figure 4 summarizes the results of the univariate feature evaluation described in Section 2.2 by aggregating feature-level effect sizes across all sixteen survival tasks. For each feature, we report the maximum absolute log-hazard ratio observed across tasks for which the feature exhibited statistically detectable association with survival time after false-discovery-rate correction. This aggregation reflects our objective of verifying that each feature contributes meaningful signal in at least one behavioral context, rather than requiring universal relevance across all tasks.

Grouping features by category reveals systematic differences in the magnitude and variability of survival associations. User-history features exhibit the largest effect sizes and greatest heterogeneity, consistent with the intuition that individual behavioral history is a primary driver of time-to-event outcomes in lending protocols. Raw transaction features and temporal features show more moderate but consistently non-negligible effects, indicating that instantaneous transaction attributes and time-of-occurrence information also encode predictive signal. Market-history features display smaller effect sizes on average, reflecting their role as contextual indicators rather than direct drivers of individual outcomes.

We emphasize that this analysis is not intended to rank individual features nor to characterize their importance within any specific predictive model. Because the benchmark models evaluated in Section 3 exploit nonlinear interactions and feature combinations, univariate associations should be interpreted as evidence of relevance rather than as proxies for multivariate importance. Instead, Figure 4 demonstrates that all feature categories contribute detectable survival signal under conservative criteria, supporting the inclusion of the full feature set in this benchmark while preserving task-specific heterogeneity.

Researchers who wish to engineer new features can do so by reconstructing user-level transaction histories from the survival data itself. The survival data does contain user-ids and timestamps for each index event. By grouping the datasets by user and sorting by timestamp, the majority of a user's transaction history can be recovered. To fully recover the transaction history for a user, the outcome events of "account liquidated" also need to be processed and recovered, which can be done by taking all of a user's index events that have "account liquidated" as their outcome event, find the events for which the outcome event was observed, and calculate the outcome's timestamp (which at this point must be a "liquidated" event) by taking the timestamp of the index event and adding to it the "timeDiff" from the survival record. These reconstructed transaction histories could then be used to design new domain-specific features, or to create a more nuanced survival dataset that accounts for competing events. Such extensions allow researchers to build upon the benchmark to explore richer temporal representations while preserving full reproducibility from public data.

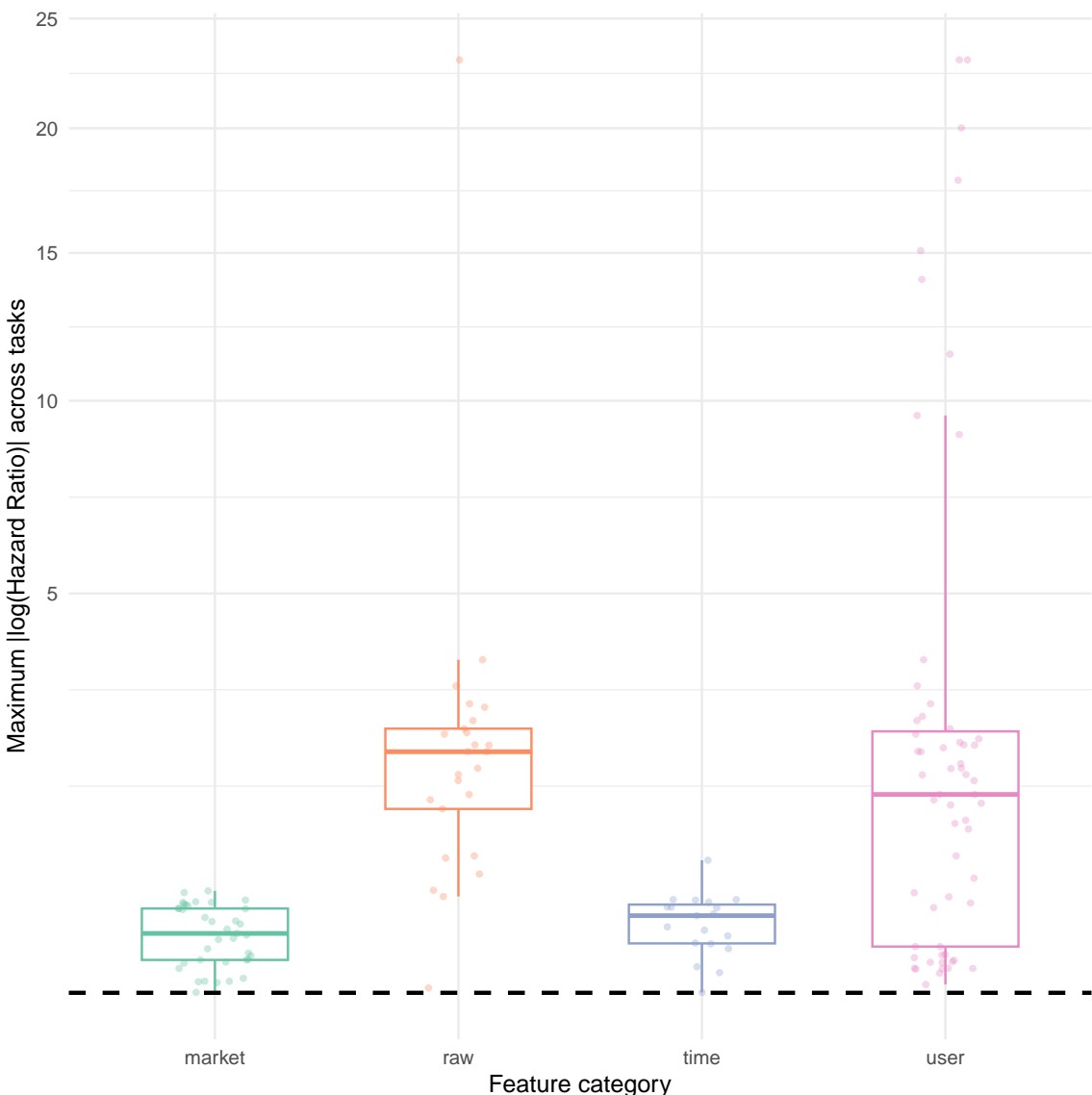

Figure 4: Distribution of the maximum absolute log-hazard ratio observed for each feature across all sixteen survival tasks, grouped by feature category. Each point corresponds to a single feature, and boxplots summarize category-level distributions. The dashed line indicates the threshold $|\log(HR)| = 0.1$ used to define non-negligible effect sizes. For visualization clarity, the y-axis is shown on a square-root scale.

Table 10: All user-level features engineered to represent a user's transaction history prior to each transaction. Features with [TYPE] are created identically for each of the five main transaction types (borrow, deposit, liquidation, repay, and withdraw).

| Feature Name | Description |
|---|---|
| userReserveMode | The most common coin used by this user. |
| userCoinTypeMode | The most common coin type (stable or non-stable) used by this user. |
| userIsNew | Whether this is the user's first transaction. |
| userSecondsSinceFirstTransaction | How many seconds have passed since this user's first transaction. |
| userSecondsSincePreviousTransaction | How many seconds have passed since this user's previous transaction. |
| userCollateralCount | How many collateral transactions this user has made in the past. |
| userSwapCount | How many Swap transactions this user has made in the past. |
| user[TYPE]Count | How many [TYPE] transactions this user has made in the past. |
| user[TYPE]Sum | The total amount of the coin involved in this transaction that this user has used in their past [TYPE] transactions. |
| user[TYPE]AvgAmount | The average amount of the coin involved in this transaction that this user has used per [TYPE] transaction. |
| user[TYPE]SumUSD | The total value, scaled to USD, of all [TYPE] transactions made by this user in the past. |
| user[TYPE]AvgAmountUSD | The average value, scaled to USD, per [TYPE] made by this user in the past. |
| user[TYPE]SumETH | The total value, scaled to Ethereum, of all [TYPE] transactions made by this user in the past. |
| user[TYPE]AvgAmountETH | The average value, scaled to Ethereum, per [TYPE] transaction made by this user in the past. |
| userActiveDaysWeekly | The number of days in the past seven days during which this user has made at least one transaction. |
| userActiveDaysMonthly | The number of days in the past 30 days during which this user has made at least one transaction. |
| userActiveDaysYearly | The number of days in the past 365 days during which this user has made at least one transaction. |

Table 11: All market-level features engineered to represent an overall market's transaction history before each transaction. Features with [TYPE] are created identically for each of the five main transaction types (borrow, deposit, liquidation, repay, and withdraw).

| Feature Name | Description |
|---|---|
| marketCollateralCount | How many collateral transactions that have been made across the whole market in the past. |
| marketSwapCount | How many Swap transactions that have been made across the whole market in the past. |
| market[TYPE]Count | How many [TYPE] transactions have been made across the whole market in the past. |
| market[TYPE]AvgAmount | The average amount of the coin involved in this transaction that users in this market have used in [TYPE] transactions in the past. |
| market[TYPE]Sum | The total amount of the coin involved in this transaction that this users in this market have used across all past [TYPE] transactions. |
| market[TYPE]AvgAmountUSD | The average value, scaled to USD, per [TYPE] made by users in this market in the past. |
| market[TYPE]SumUSD | The total value, scaled to USD, of all [TYPE] transactions made by users in this market in the past. |
| market[TYPE]AvgAmountETH | The average value, scaled to ETH, per [TYPE] made by users in this market in the past. |
| market[TYPE]SumETH | The total value, scaled to ETH, of all [TYPE] transactions made by users in this market in the past. |

Table 12: Time-based features created for each transaction in the dataset. Features prefixed with "sin[cos]" represent two separate features, one starting with "sin" and one starting with "cos".

| Feature Name | Description |
|---|---|
| timeOfDay | A numeric value between 0 and 24 representing the time of day at which the transaction took place. |
| dayOfWeek | The day of the week (1-7) during which the transaction took place. |
| dayOfMonth | The day of the month (1-31) during which the transaction took place. |
| dayOfYear | The day of the year (1-365) during which the transaction took place. |
| quarter | The quarter (1-4) during which the transaction took place. |
| dayOfQuarter | The day of the quarter (1-95) during which the transaction took place. |
| sin[cos]TimeOfDay | The time of day transformed by the sine [cosine] function. |
| sin[cos]DayOfWeek | The day of the week transformed by the sine [cosine] function. |
| sin[cos]DayOfMonth | The day of the month transformed by the sine [cosine] function. |
| sin[cos]DayOfQuarter | The day of the quarter transformed by the sine [cosine] function. |
| sin[cos]DayOfYear | The day of the year transformed by the sine [cosine] function. |
| sin[cos]Quarter | The quarter transformed by the sine [cosine] function. |
| isWeekend | A boolean flag (0 or 1) representing whether this transaction occurred on a weekend (0 if no, 1 if yes). |

