# OpenReview forum: "FinSurvival: A Suite of Large Scale Survival Modeling Tasks from Finance"
_DMLR — Accepted by DMLR_

### Review · Reviewer_JaHd · 2025-08-11

**Recommendation:** 2
**Confidence:** 3

**Summary Of Contributions:**

The authors curated a new dataset (called FinSurvival) which consists of 7.7 million records of 16 survival modeling tasks from publicly available transaction data of cryptocurrencies. In addition, the authors converted each task into a corresponding classification task and generated hand-crafted features for machine learning modeling. The authors perform benchmark evaluations of several existing statistical and machine learning methods on this dataset and discuss their performance.

**Strengths:**

Please see the Strengths And Weaknesses section above.

**Audience:**

Yes

**Broader Impact Concerns:**

As mentioned above, the authors should provide additional details and evidence that they received explicit permission from TheGraph to query the raw transaction data and curate the dataset. Besides this, I do not have other concerns over the ethical or broader impacts of this paper.

**Claims And Evidence:**

As mentioned above, the authors need to provide additional details on many aspects of this work to help validate their methodology and reported results.

**Datasets And Benchmarks:**

Additional details on data collection and processing need to be provided. Please see the Requested Changes section.

**Extended Submissions:**

This paper is not an extended version of a previously published work.

**Limitations:**

Please see the Requested Changes section.

**Requested Changes:**

1. An important domain of survival modeling is to predict the duration of clinical trials, where the ongoing clinical trials are treated as right censored and the completed trials are treated as uncensored. There are several large datasets in this domain (e.g., ClinicalTrials.gov, Citeline Trialtrove) and the authors should review these as well as the related machine learning works.

2. The authors should provide additional details and evidence that they received explicit permission from TheGraph to query the raw transaction data and release the transformed data in the curated dataset. I reviewed TheGraph website but did not find such explicit permissions automatically granted. This is a prerequisite for publication of this paper and the curated dataset.

3. The authors should provide a self-contained overview of survival modeling and restricted mean survival times (RMSTs) in the main text with sufficient technical details to help readers clearly understand how RMST is used to split the train/test datasets and create the classification tasks.

4. The authors should clearly explain whether there is any bias in the data curation process. For instance, the authors mentioned that “points that are censored in fewer than RMST days are dropped” (page 12). How many points are dropped as a result? Does this lead to any bias in the final dataset? Did the authors perform any tuning on the threshold to drop these points?

5. Additional details on feature construction (Section B.4) need to be provided. The authors should clearly explain (1) why they chose to generate this particular set of features and (2) how one may generate additional relevant features for machine learning modeling. The latter is especially important to help researchers better leverage this dataset. The authors should also cite the appropriate references to support their domain knowledge and claims that these hand-crafted features are useful for predictions.

6. Related to above, the authors should evaluate the feature importance for different models and clearly show that (1) these hand-crafted features are indeed predictive of the outcomes and (2) the feature importance results align with their domain knowledge.

7. The authors should use more recent machine learning models in survival modeling in the benchmark evaluations, especially the deep learning models reviewed in Wiegrebe et al. (2024) [1].

8. I agree with the authors that it is surprising that linear methods of logistic regression and elastic net performed the best in the classification benchmark evaluations. This may suggest that there are issues in model training (e.g., suboptimal hyperparameter tuning) for the more complex nonlinear models, which usually outperform the linear models in large-scale, high-dimensional datasets. The authors need to provide more details on model training and show that their training and hyperparameter settings are optimal.

9. In addition to SMOTE, the authors should perform ablation studies where they use different techniques to handle classification tasks with imbalanced labels (such as resampling positive and negative samples) that are more commonly used in real-world applications.

[1] Wiegrebe et al. (2024). Deep learning for survival analysis: a review. Artificial Intelligence Review. https://link.springer.com/article/10.1007/s10462-023-10681-3

**Strengths And Weaknesses:**

After reviewing the manuscript and supplementary materials, I’d recommend the authors perform a major revision and submit the manuscript for another round of review. Please find my detailed comments and questions below.

Strengths:
1. Overall this paper is clearly written and easy to understand. The technical details presented are clear and valid, although additional details need to be provided.
2. The dataset curated by the authors is a relatively novel contribution and may become a useful benchmark data for researchers in both finance and survival modeling.
3. The authors show familiarity with the recent developments in survival modeling using machine learning.
4. I thank the authors for providing the source code of their implementation on GitHub. I reviewed the code and did not find major issues.

Weaknesses:
1. The main weakness of the paper is the lack of sufficient details in many critical sections. Without these important details, it is difficult to fully validate the methodology and results reported in this paper. Please see the Requested Changes section below.
2. Currently the paper lacks a comprehensive literature review of (1) survival analysis, (2) its applications in finance and related fields, and (3) existing benchmark datasets in this field.

---

### Review · Reviewer_BKNS · 2025-08-13

**Recommendation:** 4
**Confidence:** 3

**Summary Of Contributions:**

This paper introduces a novel, large open dataset constructed from freely available DeFi data that allows for testing survival prediction models. The construction of the dataset, basic statistics and specific tasks with training and test subsets are then presented,
as well as basic benchmark models. The dataset presented here is purported to be the largest of its kind for survival modelling.

**Strengths:**

See above.
- well written paper.
- diligent statistical exploration of the dataset and careful documentation of its construction and design choices.
- clear, usable datasets allowing for simple, reproducible benchmarking.
- helpful code in R allowing for rapid development for newcomers.

**Audience:**

Yes

**Broader Impact Concerns:**

There are no broader impact concerns. The dataset is derived from readily available DeFi data.

**Claims And Evidence:**

The paper is very methodical in analysis and presenting all aspects of the dataset and challenges presented.

**Datasets And Benchmarks:**

The dataset is well documented, easily accessible and helpful code is provided. Moreover, it could easily be reconstructed from openly
available raw data and can easily be updated in future.

**Extended Submissions:**

NA

**Limitations:**

I can see no limitations of this paper.

**Requested Changes:**

I have no required changes. I would suggest addressing the few points I listed under "weaknesses" above. But even if this was not
done, this paper would not be diminished by much.

**Strengths And Weaknesses:**

Strength
- well written paper.
- diligent statistical exploration of the dataset and careful documentation of its construction and design choices.
- clear, usable datasets allowing for simple, reproducible benchmarking.
- helpful code in R allowing for rapid development for newcomers.

Weaknesses
I am happy to report that the paper has no major weaknesses. Like any piece of work, of course, one could improve it. A few examples could include:
1. Explain or rethink why a temporal cut off point (1. July 2022) was chosen between training and test data. Could a paradigm shift have occured during these two periods, specifically, did COVID19 lock-down restrictions for large parts of 2020 and 2021 not lead to different behaviour compared to the time before or even after? Maybe the training and test data should be spliced temporally? Was an analysis with the same rigor as in section 4.2 (cut-off for classification data) performed?
2. The 128 features selected stem from a prior publications of a subset of the authors. It would help the reader if the main reason for choosing these were summarized in the text.
3. One could of course go into a lengthy discussion regarding the metrics chosen and model types chosen, but a) this would never end and b) would not be helpful to this paper.

---

### Review · Reviewer_xxko · 2025-08-13

**Recommendation:** 4
**Confidence:** 3

**Summary Of Contributions:**

The authors provide a new suite of benchmark data sets for survival analysis from the financial sector (crypto currency trading).
They define a survival time as time between two subsequent events (e.g. deposit and withdrawl).
They create a separate data set for each pair of transactions, rendering 16 data sets overall. For each data set they add available and engineered features (described past transaction histories of the user and the market).
A survival task and classification task is contructed for each data set.
The data is supplemented with an initial set of benchmark results using some established learners.

**Strengths:**

The article is very well written and easy to read. The figures and tables are clear informative (minus some details).
The work is valuable as the survival community urgently needs large-scale realistic data, so the authors efforts must be commanded.
This benchmark suits also provides data from a not so common domain.
It is great that this is available open source.

**Audience:**

Yes

**Broader Impact Concerns:**

no concerns.

**Claims And Evidence:**

The empirical experiments need some attention and appear to be implausible.
Some statements are directly contradicted by the presented evidence, e.g. "important that models can learn interactions" or "FinSurvival drives methodolgical innovation in deep learning for survival modeling" , but then the DNN based methods don't perform well,

**Datasets And Benchmarks:**

There is a link to github repo, but as mentioned before, data sets + scripts should also be stored in a permanent repository.
I couldn't find the code that obtains the original data, but there is a raw data csv file in the repository.
athours did not elaborate on hosting, licensing and maintenance plan.

**Extended Submissions:**

no extended submission.

**Limitations:**

see other comments in strengs and weaknesses and Requested Changes.

**Requested Changes:**

**Section 3:**
- Please re-check your code and make sure the evaluation function is used correctly (i.e. the prediction from different learners consistently represent higher value = higher risk (or the other way round))
- More details about the XGBoost and AFT models are needed (which distribution was assumed? or was it tuned? a mixture?)
- For the machine and deep learning models, how was hyperparameter tuning performed. Which search space? What was the budget? which search strategy?
- Figure 3: The legend for the C-index is not labeled "C-Index"
- Which C-Index was used (Harrel's, Antolini's, ...), and did you avoid C-hacking: https://academic.oup.com/bioinformatics/article/38/17/4178/6640155
- I would suggest to include the (untuned) Random Survival Forest as an additional learner as it works well out of the box without tuning and can learn interactions as well as non-linearites (and deviations from PH assumption to some extent).

- I think splitting the data set by calendar time is not a good idea. The total observation period in the training data is longer. If there are features that lead to longer event times, you won't observe them in your test data. Rather, the data set should be split by user-id into train and test data. Effectively you might be introducing dependent censoring in the data by splitting it by calendar time. Also, since users can be present in both, training and test set, they are not independent.

**Section 4:**
I would suggest to remove the entire section. As explained above, this approach is not correct. If the user want's to turn it into a classification problem, they can. But I wouldn't provide them with an incorrect data set.

**Section 5:**
- "Traditional models such as XGBoost" -> in the survival context, XGBoost is not a traditional method, the AFT extension was only implemented 3-4 years ago, after the publicatoin of DeepSurv and DeepHit


** Additional remarks**
- Currently there is a link to GitHub, but this can be deleted. The data should be stored in some permanent repository, e.g. zenodo or Figshare or similar.

- I think the data should be stored in a "multi-state" type of format, i.e. start-stop notation, where for each user (and coin) there is one row for each consecutive transition, so users can construct their own suitable data set, as application of different methos and assumptions about the DGP might require different types of data pre-processing steps. In addition, there could be some functions to extract subsets of the data assuming a specific DGP (e.g. as done by the authors in order to construct the single-event data sets).

**Strengths And Weaknesses:**

**Strengths**
- The survival community is lacking realistic, high-dimensional data so the efforts by the authors
- The data provides a new and original field of application for survival analysis.
- The article is mostly well written and easy to understand.

**Weaknesses**
- Some details about the construction of the data set are not clear
- The reported results in Fig 3 seem implausible
- Construction of the classification data set is unnecessary an in my opinion not correct
- Details about the training (HPO) and evaluation set up are not given

Details:

- Data set construction: Details about data set construction are not clear. Was this done on a user-currency basis. I.e. you only looked at index-event outcome-event pairs of the same user-coin combination to construct event times? As you write, the result that the withdraw-xxx data sets are easiest to predict is quite counter-intuitive. There might be multiple withdraw-deposit pairs for the same users, so the independence assumptions will be violated?
- Section 3: The results presented in Fig. 3 seem very implausible and description of the results is inconsistent. For example, a Cox model approximates the underlying distribution quite well (especially given enough data), so if the underlying distribution is a Weibull distrubution, it should perform similarly well to the AFT Weibull model. Since both, the AFT and the Cox model are not regularised and are both proportional hazards models (I assume the authors used Weibull AFT? The information is missing from the manuscript), there shouldn't be that much difference between the models. Also, a result of .31 suggests that the model was able to lear sth from the data, so inverting the prediction would result in a quite good performance. Similarly with DeepHit and DeepSurv. At least the models should be able (with enough tuning and regularization) recover the null model (constant prediction). The persistent performance below .5 indicates that a inversion of the prediction would yield good results. I suspect the models either haven't been tuned properly, or there is a coding error. Some learners/implementation return a risk score, where higher values indicate higher risk, some learners return negative risk scores, where lower vaues indicate higher risk, some learners return survival probability predictions, where lower values indicate higher risk. Did you make sure to check the prediction outputs and to align them with the expectation for your evaluation function? Also, you write that the results indicate that learners able to learn high-dim interactions and non-linearities excel, however, the AFT and Cox are linear (in the coefficients) and DeepHit and DeepSurv are not, so the argument doesn't work.

- Section 4: I don't understand any of the motivation or justification for construction of the classification data set. You refer to discrete-time methods and RMST for justification. But RMST is a summary statistic used to describe a time-to-event distribution in a single number, but how is "This method (...) particularly valuable for high-dimensional datasets, where traditional survival models... may struggle"? You first have to get a good estimate of the Survival probability (conditional on features), then you can calculate the RMST. But the RMST itself is not a method. Also "RMST was selected as it summarizes ... without requiring the enire cohort ...". So is the survival probability at \tau, some argue it is more interpretable, as it is observed/calculated on the time-scale of interest, but that's not what you argue here. Then you use the RMST to define a cut-off to discretize the outcome (event before or after the RMST) and treat this binary outcome as target for classfication. This is wrong. Subjects censored before RMST are neither 0 nor 1. Dropping them will introduce a bias, when calculating the P(Y>RMST|x). That's why you don't perform classfication on survival tasks. The discrete-time methods you refer to define many intervals and construct a new data set where subjects are only included in an interval with 0 or 1 if they are still in the risk set at the beginning of the interval (which takes into account censoring), thus we don't estimate P(Y > RMST|x) but rather P(Y \in (lower border, upper border]|Y> lower border, x), a conditional probability. Since the intervals are small, the error made by ignoring the exact event time is small and one can effectively approximate the event-time distribution at different time-points. Not so in your approach. You can still turn this into a classification problem, but you have to use the time and censoring information to construct IPCW weights (see Vock et al, https://www.sciencedirect.com/science/article/pii/S1532046416000496). This also implies that you will obtain a prediction of S(RMST|x) and need to use survival metrics to evaluate the predictions, as the test set will also contain censoring.